# REVISITING FEW-SHOT OBJECT DETECTION WITH VISION-LANGUAGE MODELS

## ABSTRACT

Few-shot object detection (FSOD) benchmarks have advanced techniques for detecting new categories using limited annotations. Existing FSOD benchmarks repurpose well-established datasets like COCO by partitioning categories into `base` and `novel` classes for pre-training and fine-tuning respectively. However, these benchmarks do not reflect how FSOD is deployed in practice. Rather than pre-training on only a small number of categories, we argue that it is more practical to download a foundational model (e.g., a vision-language model (VLM) pretrained on web-scale data) and finetune it for specific applications. Surprisingly, we find that zero-shot inference from foundational VLMs like GroundingDINO significantly outperform state-of-the-art methods (48.3 vs. 33.1 AP) on COCO, suggesting that few-shot detection should be reframed in the context of foundation models. In this work, we propose a new FSOD benchmark protocol that evaluates detectors pre-trained on any external dataset (not including the target dataset), and finetuned on $K$-shot annotations per $C$ target classes. Further, we note that FSOD benchmarks are actually federated datasets, which are exhaustively annotated for a single category only on a subset of data. We leverage this insight and propose simple strategies for fine-tuning VLMs to improve FSOD. We demonstrate the effectiveness of our approach on LVIS and nuImages.

## 1    INTRODUCTION

Object detection is a fundamental problem in computer vision (Felzenszwalb et al., 2009; Lin et al., 2014) that has matured in recent years (Ren et al., 2015; Liu et al., 2016; Redmon & Farhadi, 2017; Lin et al., 2017). Given a large-scale annotated dataset, one can easily train a detector from scratch. However, training object detectors for domains with limited annotated data remains challenging, motivating the problem of few-shot object detection.

**Status Quo.** Few-shot object detection (FSOD) benchmarks have made considerable progress on learning to detect new categories from limited training data. Existing FSOD benchmarks are constructed by partitioning popular object detection datasets like PASCAL VOC (Everingham et al., 2010) and COCO (Lin et al., 2014) into `base` categories (with many examples per class) and `novel` categories (with few examples per class). Detectors are first trained on `base` classes to learn a strong region proposal network (RPN) and are then finetuned on $K$ examples (or $K$-shots) from both `base` and `novel` classes. The goal is to detect `novel` categories from only a few training examples while maintaining performance on `base` classes. Performance (measured by average precision) is reported across both `base` and `novel` classes.

Historically, concept leakage (e.g. shared classes) between the pre-training and fine-tuning steps has been of significant concern (Hsieh et al., 2019; Zhu et al., 2021; Chen et al., 2021; Köhler et al., 2021). FSOD benchmarks carefully construct dataset splits such that `base` and `novel` classes are disjoint. However, as most detectors are initialized with ImageNet (Deng et al., 2009) weights, concept leakage of common categories in COCO and PASCAL VOC already occurs. For example, `cup` and `person` are present in both ImageNet and COCO. Similarly, although the COCO class `car` is not present in ImageNet, similar concepts like `sports car` and `race car` are included.

**Technical Insights.** We argue that allowing vocabulary overlap between pre-training and `novel` classes (as is the case with foundational vision models) is more practical. Modern practitioners simply download the latest state-of-the-art detector pre-trained on large-scale datasets (which may

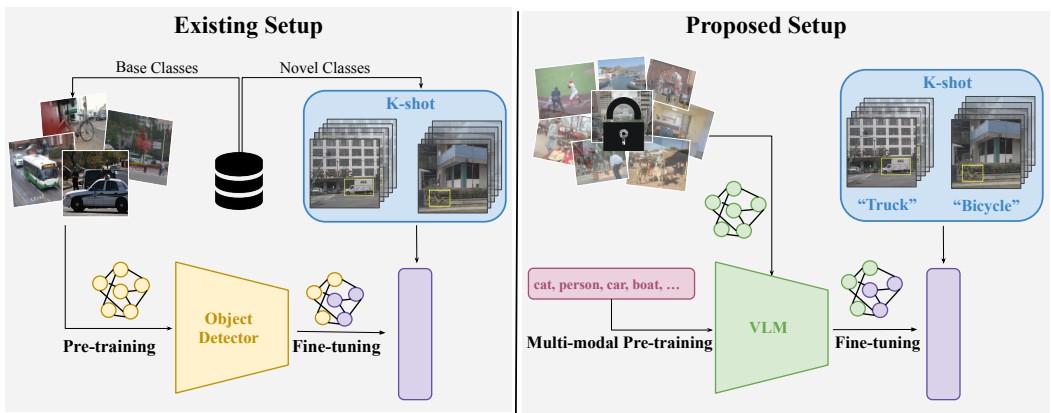

Figure 1: **Proposed FSOD Setup.** We propose a new setup for FSOD which embraces Foundational VLMs. On the **left**, we describe the existing setup: Given a base and novel class split, FOSD methods pre-train on the base classes and then finetune on $K$-shots of the novel (and optionally base) classes. On the **right** is our proposed setup: Given the scale and often private nature of data used to train VLMs, it is impractical to maintain a split of base and novel classes. Instead, one should directly fine-tune VLMs on $K$-shots of the target classes (and evaluate only those classes). Importantly, VLMs allow us to exploit additional language cues (such as class names and descriptions) for fine-tuning. We show that such "zero-shot" language cues without any $K$-shot fine-tuning already outperforms state-of-the-art FSOD methods.

not be publicly available) (Shao et al., 2019a; Kebe et al., 2021; Li et al., 2022; Sharma et al., 2018; Radford et al., 2021) and finetune it for their task. Recent work (Li et al., 2022; Liu et al., 2023; Zhou et al., 2022) also advocates for this more flexible setting, allowing vocabulary overlap between training and test sets.

Under this realistic setup, state-of-the-art vision-language models (VLMs) like GroundingDINO achieve 13.6% higher AP than leading FSOD methods on COCO (48.3 vs. 33.1) *without fine-tuning* (cf. Table 1). Importantly, (1) zero-shot inference with VLMs serves as a strong baseline for few-shot object detection, and (2) ignoring this realistic setup impoverishes research exploration. Current benchmarks should be amended to more accurately reflect practical applications.

Somewhat surprisingly, naively fine-tuning the last few layers of a VLM in the $K$-shot setup does not always improve performance over zero-shot inference because training images are not exhaustively labeled (Fig 2). Importantly, objects not annotated in the frame are considered negatives during training. We find that improperly training on sparsely annotated images yields degraded performance. However, we find that FSOD benchmarks are actually federated datasets (Gupta et al., 2019). A federated dataset is a single dataset comprised of smaller subsets, where each subset guarantees exhaustive annotations for a single category. Inspired by prior work in learning with federated datasets (Zhou et al., 2021), we demonstrate that training VLMs with federated losses for FSOD consistently improves over zero-shot inference (cf. Tables 2, 4).

**Contributions.** We present three major contributions

1. We amend few-shot object detection benchmarks to more closely align with practical applications by not artificially restricting concept leakage during pre-training
2. We point out that existing FSOD benchmarks are actually federated datasets, and present simple strategies for fine-tuning VLMs
3. We conduct extensive experiments to ablate our design choices and demonstrate that our simple method achieves state-of-the-art results on LVIS and nuImages FSOD benchmarks

## 2 RELATED WORKS

**Few-Shot Object Detection** aims to detect new object categories given limited training data (Köhler et al., 2021). Recent work explore two primary approaches: meta-learning and transfer learning. Meta-learning-based methods focus on acquiring generalizable features from a set of base classes,

which can then be applied to detect objects in novel classes. For example, Kang et al. (2019) proposed a technique that re-weights features from base classes to predict novel classes. Xiao et al. (2022) defines a simple yet effective framework addressing both few-shot object detection and few-shot viewpoint estimation. Fan et al. (2020) introduced a general few-shot object detection network that learns a matching metric between image pairs, while (Wu et al., 2021) enhanced object features using a universal prototype. In contrast, transfer learning involves freezing the network weights pretrained on a `base` dataset to improve the model's ability to generalize to novel classes with limited data. Transfer learning approaches often follow a two-stage fine-tuning strategy: first training on the `base` dataset and then fine-tuning only the box classifier and regressor with `novel` data. This strategy, as demonstrated by Wang et al. (2020), has proven to outperform previous meta-learning approaches. Recent work has primarily focused on improving classification performance. FSCE (Sun et al., 2021) utilizes a contrastive proposal encoding loss to encourage instance-level intra-class compactness and inter-class variance. Similarly, Li et al. (2021) applied a class margin loss technique to balance inter and intra-class margins. Our approach leverages transfer-learning by fine-tuning vision-language models (VLMs) pre-trained on large-scale datasets.

**Vision Language Models** are trained using weakly-supervised image-text pairs collected from the web. These models embed images and text into a shared space, enabling open-vocabulary detection. Early work adapted VLMs for object detection by either distilled the model's predictions for specific image regions (Gu et al., 2021) or directly incorporated detection components into the frozen Kuo et al. (2022) or finetuned (Minderer et al., 2022; 2023) encoders. In contrast, RegionCLIP (Zhong et al., 2022) employs a multistage training approach, which involves generating pseudo-labels from captioning data and then conducting region-text contrastive pretraining before transferring to detection. GLIP (Li et al., 2022) uses a single text query for the entire image and frames detection as the problem of phrase grounding . Detic (Zhou et al., 2022) addresses long-tail detection performance by leveraging image-level supervision. In the context of open-vocabulary detection, there might be overlap between the object categories seen during training and those encountered during testing. We use the term zero-shot inference to identify that a model has never been trained on the target dataset.

**Federated Datasets** are constructed by combining numerous smaller datasets, each resembling a conventional object detection dataset for a single category (Gupta et al., 2019). Each of these smaller datasets ensures exhaustive annotations for a specific category. Images within each smaller dataset may overlap, resulting in some images with exhaustive annotations for multiple categories. Importantly, since exhaustive annotations for a particular category are only guaranteed within each small dataset, most images are sparsely annotated. Consequently, naively training models with federated datasets leads to much sparser gradients, particularly for infrequently occurring classes. To address this challenge, CenterNet2 (Zhou et al., 2021) introduced FedLoss, a simple modification of cross-entropy loss which randomly samples a subset of negative categories for each image. We adopt FedLoss for Few-Shot Object Detection (FSOD) and find that it consistently improves fine-tuning performance.

## 3 FEW-SHOT OBJECT DETECTION WITH VISION LANGUAGE MODELS

As shown in Fig 1, our proposed few-shot object detection (FSOD) protocol uses vision-language models (VLMs) pre-trained on diverse, large-scale datasets prior to fine-tuning on $K$-shots per $C$ target classes. We contrast our proposed setup with the standard FSOD benchmark, demonstrate that FSOD benchmarks are actually federated datasets, and present simple strategies for fine-tuning VLMs below.

### 3.1 FSOD BENCHMARKS MUST BE RE-FRAMED IN THE CONTEXT OF FOUNDATIONAL VLMS

Existing FSOD benchmarks re-purpose well-established datasets like PASCAL VOC (Everingham et al., 2010) and COCO (Lin et al., 2014) by partitioning them into `base` classes for pre-training and `novel` classes for fine-tuning. For COCO, the 60 categories disjoint with PASCAL VOC are used as `base` classes while the remaining 20 classes are used as `novel` classes (Wang et al., 2020). However, we argue that this setup is artificial, as it requires FSOD methods to detect `car` and `person`, among many other common categories given few-shot examples. Importantly, VLMs like

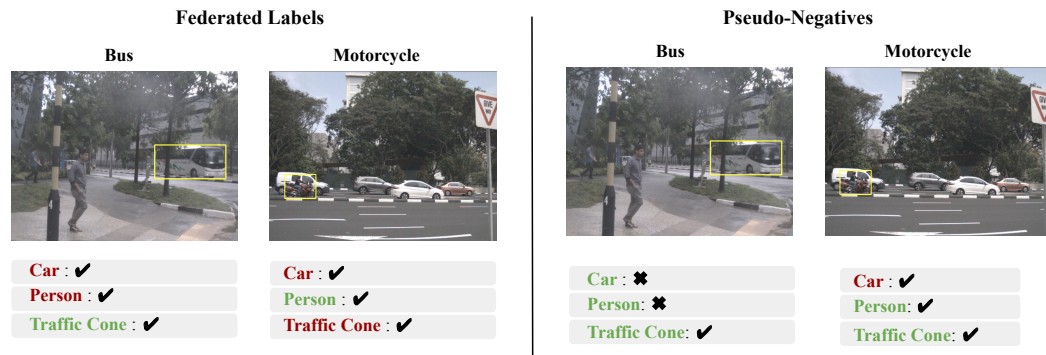

Figure 2: **Federated Labels vs Pseudo-Negatives**. The **left** visualizes the standard $K$-shot detection setup, which we argue is actually a *federated* dataset (Gupta et al., 2019) where one is given multiple mini-datasets of $K$-images. In this case, we visualize two $K = 1$ datasets of `buses` and `motorcycles`. Importantly each mini-dataset does *not* provide information about the presence of other objects. Previous FSOD methods apparantly ignore this fact, and instead assume the collective set of few-shot images are *fully* annotated across all object classes. This will likely produce many incorrect negative labels – e.g., all unlabeled `car`'s in the background of the `motorcycle` mini-dataset will be incorrectly treated as negative cars. We use a ✓to denote that a given image will be treated as a negative example of a given class by the learner and a ✗to denote that a given image will be ignored when learning a given class. We color such negative labels as green when correct and red when incorrect. Naive FSOD approaches learn about *all* classes from *all* images, which results in many incorrect negative labels (shown in red on the **left**). Instead, we embrace the partially-labeled nature of the data and exploit tools from weakly-supervised learning, such as the use of psuedo-labels predicted by a teacher. We train (or rather, fine-tune) initial detectors on only the appropriate mini-dataset and use thresholded psuedo-detections to find images that can be confidently treated as (pseudo) negatives, which results in much fewer mistakes (shown in red on the **right**). This in turns produces improved performance. We also attempted to learn from psuedo positive labels, but found these to be less reliable.

GroundingDINO can already detect common categories with high accuracy *without fine-tuning* on COCO (cf. Table 1).

Although Wang et al. (2020) highlights the importance of evaluating `base` class accuracy to prevent catastrophic forgetting, many foundational vision models are trained on large-scale private datasets. For example, CLIP (Radford et al., 2021) pre-trained weights are freely available, but the original dataset of 400M image-text pairs has not been publicly released. In the context of foundational models, we argue that partitioning datasets into `base` and `novel` class no longer makes sense. Instead, FSOD methods should only train on $K$-shot annotations for $C$ target classes, and also evaluate performance on these $C$ classes. We demonstrate this more realistic setup with nuImages in Section 4.4.

## 3.2 FSOD BENCHMARKS ARE FEDERATED DATASETS

Prior works follow the $K$-shot dataset creation process established by Wang et al. (2020). To construct a $K$-shot dataset $D$, select an image $I$ and class $C$. If the total annotations in $I$ for class $C \leq K$, then we add $I$ to $D$. We repeat this process for all classes utill we have exactly $K$ annotations per class. Similarly, a federated dataset is comprised of smaller subsets, where each subset guarantees exhaustive annotations for a single category. This hints that we can leverage insights about federated datasets and apply them to FSOD tasks.

## 3.3 FINE-TUNING VLMS FOR FSOD

Although VLMs achieve strong zero-shot performance on common classes (as demonstrated by GroundingDINO's accuracy on the COCO FSOD benchmark), they struggle to detect ambiguously defined classes like `trailer`. nuImages (Caesar et al., 2020) defines `trailer` as independent from the truck cab. However, Detic jointly detects the truck cab and its trailer together (cf. Figure 3). This fine-grained distinction is provided to the annotators in the form of visual examples. Not providing such visual examples or detailed category definitions would lead to poor inter-annotator

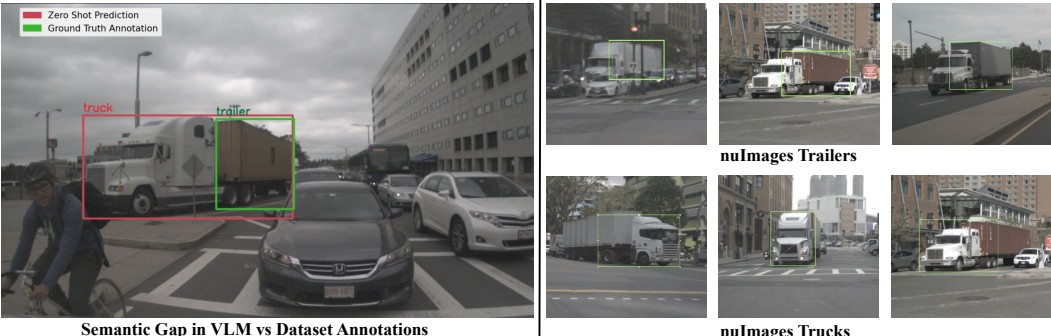

**Semantic Gap in VLM vs Dataset Annotations**

**nuImages Trailers**

**nuImages Trucks**

Figure 3: **Knowledge Gap between VLM Pre-training and Dataset Annotations**. While VLMs show impressive zero-shot performance, they struggle when the concept of the target class is different from vision-language concepts encountered in web-scale training. On the **left**, we see that the nuImages dataset defines the cab of the truck as a separate object concept from its trailer (shown in green), while the zero-shot VLM predicts the entire vehicle as a truck (shown in red). The **right** visualizes the actual *class definitions* given to nuImage annotators, provided as both textual descriptions and visual examples. Just as human annotators learn concepts from such few-shot multimodal vision-language data, VLM fine-tuning similarly exploits such multimodal language and visual cues for few-shot learning.

agreement for certain classes. Similarly, to align the VLMs with dataset annotations, we provide a few exemplars for each class. Specifically, we suggest multi-modal fine-tuning of VLMs to address these semantic knowledge gaps.

We start by fine-tuning Detic (Zhou et al., 2022) on the provided $K$-shot examples. We ablate the impact of freezing different parts of Detic in Table 5 and find that freezing the backbone, RPN, and classifier head with CLIP embeddings yield the best performance.

Due to the sparse annotation nature of the FSOD task, we posit that the model will receive sparser gradients which degrades the object detector's performance, especially for rare classes. This is because all unannotated objects in the image would be treated as negatives(Tan et al., 2020). Therefore, we explore three strategies for handling negatives.

We fine-tune Detic with Federated Loss (FedLoss) (Zhou et al., 2021) using a subset $S$ of classes for each training image. Specifically, we use a binary cross-entropy loss on all classes in $S$ and ignore classes outside of $S$ during training. $S$ comprises of the ground-truth annotation class along with randomly sampled negative classes for each image. We sample these negative classes in proportion to their square-root frequency in the training set. We find that probablistically sampling negatives rather than labeling all unannotated classes as negatives improves fine-tuning results, reliably beating zero-shot performance.

However, we note that FedLoss samples common classes like car more frequently as negative examples, hurting detection accuracy for long-tailed datasets like LVIS and nuImages. Instead, we propose Inverse FedLoss (InvFedLoss), a simple modification of FedLoss that samples negative categories in proportion to the *inverse* of their square root frequency. This ensures that we sample rare categories as negatives more frequently to better match the true data distribution. Importantly, although we only train with $K$ examples per class, we evaluate on a test-set which is exhaustively annotated. Leveraging this insight improves over FedLoss and naive fine-tuning.

Despite the effectiveness of Inverse FedLoss, probablistically sampling negatives using dataset-wide statistics is sub-optimal because it does not consider the content of each image. We can improve the correctness of sampled negatives by using pseudo-labels to determine which classes are likely not in a particular image. Specifically, we first finetune Detic using Inverse Federated Loss and predict per-image pseudo-labels. If the maximal score for any class prediction is less than a fixed threshold, we consider this class to be a negative. Using image predictions to identify pseudo-negatives yields better results than simply using dataset-wide statistics.

| Approach | 30-shots | | |
|---|---|---|---|
| | AP | bAP | nAP |
| FRCN-ft-full (Yan et al., 2019) | 18.6 | 20.6 | 12.5 |
| FRCN-BCE (Yan et al., 2019) | 30.2 | 36.8 | 10.3 |
| TFA w/ fc (Wang et al., 2020) | 29.3 | 34.5 | 13.5 |
| TFA w/cos (Wang et al., 2020) | 29.9 | 35.3 | 13.6 |
| MPSR (Wu et al., 2020) | 17.1 | 18.1 | 14.1 |
| Meta-RCNN (Yan et al., 2019) | 7.8 | 7.1 | 9.1 |
| FsDetView (Xiao et al., 2022) | 10.0 | 9.3 | 12.0 |
| Retentive R-CNN (Fan et al., 2021) | 32.9 | 39.3 | 13.8 |
| DiGeo (Ma et al., 2023) | 33.1 | 39.4 | 14.2 |
| **GroundingDINO (Zero-Shot)** (Liu et al., 2023) | **48.3** | **46.3** | **54.3** |

Table 1: **COCO Few-Shot Object Detection Performance.** Zero-Shot inference with VLMs like GroundingDINO easily surpass the performance of state-of-the-art FSOD methods, motivating the need to re-frame FSOD in the context of foundational VLMs. Although GroundingDINO has not seen COCO images during training, it has seen examples of COCO classes.

## 4    EXPERIMENTS

In this section, we highlight that zero-shot inference from VLMs significantly improves over state-of-the-art FSOD approaches, suggesting that existing benchmarks should be re-framed in the context of foundational vision models. In addition, we demonstrate the federated losses improves few-shot object detection under both the standard and our proposed FSOD setups. Lastly, we ablate the impact of freezing different detector components on fine-tuning performance. We will release our code to facilitate future research in FSOD.

### 4.1    DATASETS AND METRICS

We re-purpose three established datasets for few-shot object detection, described below.

- **COCO** (Lin et al., 2014) is a well-established dataset with 80 classes. 60 categories disjoint with PASCAL VOC are used as `base` classes ($C_b$) while the remaining 20 classes are used as `novel` classes ($C_n$) (Wang et al., 2020). We evaluate methods by reporting average precision on $C_b$ (bAP), $C_n$ (nAP) and $C_b \cup C_n$ (AP).

- **LVIS** (Gupta et al., 2019) re-annotates COCO images using 1,230 fine-grained classes, which are divided into frequent, common and rare based on the cardinality of each class. Frequent and common classes are combined to form `LVIS-base` and is used for pre-training. Rare classes are used for `LVIS-rare`. Following Wang et al. (2020); Ma et al. (2023), we report performance across the groups ($AP_f, AP_c, AP_r$) on the LVIS val-set.

- **nuImages** (Caesar et al., 2020) annotates 18 classes, which are divided into classes with `many`, `medium`, and `few` examples (Peri et al., 2023). Although not traditionally used for few-shot object detection, nuImage's open-world categories like `debris` and `pushable-pullable` make it particularly challenging.

### 4.2    ZERO SHOT INFERENCE BEATS SOTA FSOD METHODS

We compare state-of-the-art FSOD methods with zero-shot inference from GroundingDINO Liu et al. (2023) on COCO in Table 1. Surprisingly, GroundingDINO beats DiGeo (Ma et al., 2023) by 16.2% AP averaged across both `base` and `novel` categories despite never training on COCO images. GroundingDINO's impressive performance is due to its large-scale multi-modal pre-training on Objects365 (Shao et al., 2019b), GoldG and Cap4M (Li et al., 2022). High accuracy on `novel` classes (which include common categories like `car`, `person`, `dog`, `cat`) further highlights limitations of the standard FSOD setup. Existing benchmarks must be re-framed in the context of foundational vision models.

| Approach | 10-shots | | | |
|---|---|---|---|---|
| | $AP$ | $AP_f$ | $AP_c$ | $AP_r$ |
| TFA w/ fc (Wang et al., 2020) | 24.1 | 27.9 | 23.9 | 14.9 |
| TFA w/ cos (Wang et al., 2020) | 24.4 | 27.7 | 24.3 | 16.9 |
| DiGeo (Ma et al., 2023) | 24.9 | 28.5 | 24.6 | 17.3 |
| Detic (`Base Only`) (Zhou et al., 2022) | 30.0 | 34.4 | 30.8 | 16.3 |
| + Fine-Tuning (`Base` + `Novel`) | 30.0 | 33.2 | 31.9 | 15.5 |
| w/ FedLoss | 30.8 | 33.9 | 32.7 | 17.4 |
| w/ InvFedloss | 31.1 | 34.3 | 32.5 | 18.7 |
| w/ Pseudo-Negatives | **31.6** | **34.6** | **33.2** | **19.2** |

Table 2: **LVIS Few-Shot Object Detection Performance.** We evaluate the impact of federated losses independent of large-scale pre-training. Therefore, we follow the standard FSOD setup and pre-train Detic from scratch on `LVIS-base`. We note that Detic em without fine-tuning on `LVIS-rare` outperforms all prior approaches. Using our insight that FSOD benchmarks are actually federated datasets helps us improve performance by $\sim 2$ points on $AP_r$ over standard fine-tuning. We use $|S| = 50$ for FedLoss and InvFedLoss experiments, where S is the set of sampled classes. Training with federated loss improves fine-tuning performance because we don't naively assume all classes not labeled in an image are negatives.

## 4.3 EVALUATING LVIS UNDER THE STANDARD FSOD SETUP

Recall that FSOD benchmarks are actually federated datasets. To evaluate this claim independently of large-scale pre-trained vision language models, we train Detic (Zhou et al., 2022) (since GroundingDINO does not provide training code) from scratch on `LVIS-base`. We use a ResNet-50 backbone for fair comparison with prior work (Wang et al., 2020; Ma et al., 2023).

As shown in Table 2, Detic outperforms all other baselines without fine-tuning on `LVIS-rare` due to its multi-modal training. Impressively, Detic beats DiGeo (Ma et al., 2023)by about $\sim 6$ points on $AP_c$ and $AP_f$ and achieves $16.3$ $AP_r$ without ever seeing any rare class data. Further, fine-tuning using our insights about federated datasets further improves rare class performance by 1.8% over naive fine-tuning. Despite not pre-training on large-scale datasets, vision-language models provide considerable improvement over prior work. Specifically, Detic's CLIP-based classifier dramatically improves few-shot `novel` class detection.

## 4.4 EVALUATING NUIMAGES UNDER OUR PROPOSED FSOD SETUP

In the context of foundational models, we argue that partitioning datasets into `base` and `novel` classes no longer makes sense. Instead, FSOD methods should only train on $K$-shot annotations for $C$ target classes, and also evaluate performance on these $C$ classes. We highlight the performance of our best model under this setup in Table 3. Since the quality of samples in the few-shot split can have a significant impact on the overall performance, we run each K-shot experiment over 3 random data splits and report the average. As one would expect, detection accuracy improves as we add more training examples. Despite large-scale pre-training, we see low accuracy for classes with `few` examples, highlighting the difficulty of the nuImages datasets.

High intra-class variance for categories such as `debris` makes it difficult to generalize given few examples. Acording to nuImage's annotation instructions, `debris` can include anything that is too big to be safely driven over. This includes things like *fallen tree branch* and *trash bags*. Similarly `pushable-pullable` is an open-world category that is difficult to classify with few examples and includes *trash cans, luggage, dollies, wheel barrows, shopping carts*.

To further contextualize our results, we compute upper bounds when given access to ground truth negatives and exhaustive annotations. To compute the set of ground-truth negatives, we use the exhaustive ground-truth annotations to determine which categories are not present for each image. Note that this information doesn't exist in LVIS because its ground-truth is sparsely annotated. Training with ground-truth negatives provides an upper bound on our pseudo-negatives experiment. Next, we train using exhaustive ground-truth annotations to provide an upper bound for the specific set of images used during training. In addition, this experiment highlights the performance gap between having exhaustive negatives and exhaustive annotations.

| Approach | Average Precision (AP) | | | |
| --- | --- | --- | --- | --- |
| | All | Many | Medium | Few |
| Detic (Zero-Shot) (Zhou et al., 2022) | 14.26 | 27.28 | 16.88 | 2.36 |
| Ours (5-shots) | 15.61 | 28.88 | 18.59 | 3.12 |
| Ours (10-shots) | 16.17 | 29.61 | 19.76 | 2.91 |
| Ours (30-shots) | 17.20 | 30.48 | 21.60 | 3.37 |

Table 3: **nuImages Few-Shot Object Detection Performance.** We repurpose nuImages for FSOD by following the dataset creation process established by Wang et al. (2020).We group categories by frequency into classes with `many`, `medium` and `few` examples following Peri et al. (2023). We evaluate 3 different sets of $K$ examples and report the average accuracy. Unsurprisingly, detection accuracy improves as we add more training examples. Interestingly, accuracy across cardinalities shows a decreasing trend despite all classes being trained with $K$ examples. This suggests that despite pre-training on web-scale datasets, VLMs still struggle to detect rare categories. We note that scores struggle to improve or classes with `few` examples, highlighting the challenge of working with nuImages.

Table 4 shows that using pseudo-negatives nearly matches the true negative upper bound (16.17 AP vs 16.71 AP). This demonstrates that we are able to reliably estimate negatives in an image, alleviating the problem of learning with sparse annotations. Training with exhaustive annotations yields significantly better results for classes the `many` and `medium` examples. This is unsurprising because 10-shot FSOD includes 10 car annotations and exhaustively annotating the same images includes over 550 car annotations!

Despite strong performance on classes with `many` and `medium` examples, the upper bound for classes with `few` examples remains low (3.64 AP and 3.25 AP). We posit that it is very hard to capture the correct semantics of nuImages' rare categories only using $K$-shots. We observe similar trends for the 5 and 30-shot cases and present further analysis in the supplement.

Given the success of training with pseudo-negatives, a natural next-step is to train with pseudo-positives. Our preliminary results suggest that incorporating pseudo-positives does not provide significant improvement over simply trainign with pseduo-negatives. We posit that training with incorrect pseudo-positives may incur a higher penalty than training with incorrect psuedo-negatives. This is a promising direction for future work.

| Approach | **10 Shots**: Average Precision (AP) | | | |
| --- | --- | --- | --- | --- |
| | All | Many | Medium | Few |
| Detic (Zero-Shot) (Zhou et al., 2022) | 14.26 | 27.28 | 16.88 | 2.36 |
| + Fine-Tuning | 15.39 | 26.94 | 19.31 | **3.30** |
| w/ FedLoss | 15.47 | 27.81 | 19.69 | 2.37 |
| w/ Inverse FedLoss | 15.50 | 27.85 | 19.48 | 2.73 |
| w/ Pseudo-Negatives | **16.17** | **29.61** | **19.76** | 2.91 |
| w/ True Negatives | 16.71 | 29.58 | 20.51 | 3.64 |
| w/ Exhaustive Annotations | 18.13 | 33.53 | 21.84 | 3.25 |

Table 4: **Analysis of nuImages 10-shot Performance**. We compare the accuracy of our proposed approach against upper bounds computed for the FSOD task. Our pseudo-negatives strategy approaches the performance of using ground-truth negatives, showing that pesudo-labels can provide a reliable signal about negatives, especially across classes with `many` and `medium` examples. The performance gap between our best method and exhaustive annotations can be attributed to the large number of extra annotations, particularly for classes with `many` and `medium` examples.

## 4.5 ABLATION ON FINE-TUNING DETIC

We explore different fine-tuning strategies for training Detic with few-shot annotations. We broadly divide Detic's architecture into four components: Backbone, Region Proposal Network (RPN), Box Regressor, and Classifier. We ablate the impact of freezing different components and present results in Table 5.

Intuitively, since we only have limited training data, we attempt to fine-tune a minimal number of parameters. As shown in Table 5 initializing and freezing the classifier head with CLIP embeddings corresponding to class names provides the most significant improvement. Prior works that fine-tune vision-only models have no notion of language embeddings and therefore train must classifiers from scratch. In contrast, Detic can represent any concept using CLIP embeddings and can more easily adapt using few examples.

We find that the best configuration is to freeze the backbone, RPN and classifier head with CLIP embeddings, and simply train the classifier projection layer and box regressor. This intuitively makes sense as the pretrained Detic model has been trained on a large corpus of data. As a result, the RPN can easily pick up objects in new images and doesn't need to specifically adapt to new datasets.

| Detic Components | | | | 10 Shots: Average Precision (AP) | | | |
|---|---|---|---|---|---|---|---|
| Backbone | RPN | Box Regressor | Classifier | All | Many | Medium | Few |
| ✗ | ✗ | ✗ | ✗ | 12.11 | 19.41 | 18.44 | 0.87 |
| ✓ | ✗ | ✗ | ✗ | 12.37 | 21.20 | 17.66 | 0.91 |
| ✓ | ✗ | ✗ | ✓ | 15.08 | 22.88 | **20.99** | **3.78** |
| ✓ | ✓ | ✗ | ✗ | 11.63 | 21.65 | 15.42 | 0.83 |
| ✓ | ✓ | ✗ | ✓ | **15.37** | **26.93** | 19.73 | 2.83 |
| ✓ | ✓ | ✓ | ✗ | 10.66 | 18.54 | 15.53 | 0.56 |
| ✓ | ✓ | ✓ | ✓ | 15.31 | 26.83 | 19.58 | 2.89 |

Table 5: **Detic Fine-Tuning Ablation**. ✓ means that we freeze that component, and ✗ means we update its parameters in fine-tuning. The Detic classifier consists of a fully connected projection layer followed by a classifier head. ✓ for the classifier means that we freeze the CLIP embeddings for the classifier head and only training the classifier projection layer. Therefore, ✗ denotes not using CLIP embeddings as classifier head, and training both the projection and classifier head. We find that freezing the backbone and RPN and initializing the classifier head with CLIP embeddings yields the highest overall accuracy.

## 5 CONCLUSION

In this paper, we revisit few-shot object detection (FSOD) with vision-language models (VLMs) and find that zero-shot inference from state-of-the-art VLMs like GroundingDINO significantly outperform leading FSOD methods on COCO. We argue that existing benchmarks should be amended to include foundation models pre-trained on (often private) web-scale datasets, more closely aligning with practical applications of FSOD. In addition, we point out that FSOD benchmarks are actually federated datasets, and demonstrate that federated losses improve FSOD fine-tuning performance.

**Limitations.** Despite using a VLM pre-trained on large-scale datasets, we find that performance for rare categories (as defined by the cardinality of each class in the original dataset) is considerably lower than for common classes. We posit that VLMs are pre-trained with imbalanced data which includes many examples of common categories like `truck` and few examples of rare categories like `stroller`.

**Future Work.** Few-shot object detection with VLMs, particularly for rare categories, remains challenging and requires further investigation by the community. Interestingly, VLMs like Detic (Zhou et al., 2022), GLIP Li et al. (2022), and GroundingDINO (Liu et al., 2023) are trained with different data sources, leading to dramatically different zero-shot performance on novel categories like `stroller`. Ensembling predictions from different VLMs may yield better detection accuracy for rare categories.

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
