# OpenReview forum: "Revisiting Few-Shot Object Detection using Vision-Language Models"
_ICLR.cc/2024/Conference — ICLR 2024 Conference Withdrawn Submission_

### Official Review · Reviewer_sSNJ · 2023-10-27

**Soundness:** 2 fair
**Presentation:** 2 fair
**Contribution:** 2 fair
**Rating:** 5
**Confidence:** 4

**Summary:**

This article discusses a new benchmark of VLM for small sample target detection. The authors analyze the limitations of existing FSOD methods in practical applications and propose new applications using VLM in FSOD settings. The author proposed that the FSOD method is highly related to the federated learning method, and proposed InvFedLoss. Experimental results show that the proposed method can effectively improve the baseline method Detic.

**Strengths:**

- This paper points out many shortcomings of the traditional FSOD method, such as concept leakage caused by pre-trained models trained on ImageNet, and the training data is not fully labeled with categories.
- The author relates the FSOD problem to the federated learning problem and refers to the solution of federated learning.
- Provides a new benchmark for FSOD tasks.

**Weaknesses:**

Major:

- The method part of the article is mainly narrative. It would be clearer if the loss function could be expressed with a formula. Such as *InvFedLoss*. The clarity of the article could be improved.
- The full text, including supplementary materials, does not have visualizations and cannot intuitively demonstrate the advantages of this method.
- The methods compared in this article have no advantages. Why are Tables 2 to 4 no longer comparing Grounding DINO? Even DiGeo, which targets generalized small sample detection tasks, is not a challenging method in 23 years. It is recommended that the author refer to [1].
- Why does the experimental benchmark not include the MS-COCO dataset? The author mentioned the COCO dataset in Section 4.1.
- The author only improved InvFedLoss, and most other technologies are already available. I think the author should propose another improvement specific to this benchmark to increase the innovation of this article.
- This article should use some existing open-vocabulary object detection as a baseline. Even if it does not use the latest methods, the author should at least migrate some classic methods, such as at least one method from ViLD [2], DetPro [3], RegionCLIP [4], etc., and then use it in Few-shot Fine-tune under the settings and observe the defects of these methods under this benchmark (originally for zero-shot settings). Or demonstrate the plug-and-play capabilities of the method proposed in this article on these methods.

Minor:

- Figure 1 can be beautified to make it more compact. The font size in the picture is equivalent to the font size in the caption.
- For section 3.1, it is not recommended to give a name that is too long. It is recommended to simplify it, such as problem formulation.
- The clarity of the article could be improved.

Suggestion:

- A paper on open-vocabulary object detection also shows experiments in a few-shot setting [5]. The author can take a look.

[1] Xu, Jingyi, Hieu Le, and Dimitris Samaras. "Generating Features with Increased Crop-related Diversity for Few-Shot Object Detection." *CVPR*. 2023.

[2] Gu, Xiuye, et al. "Open-vocabulary Object Detection via Vision and Language Knowledge Distillation." *ICLR*. 2021.

[3] Du, Yu, et al. "Learning to prompt for open-vocabulary object detection with vision-language model." *CVPR*. 2022.

[4] Zhong, Yiwu, et al. "Regionclip: Region-based language-image pretraining." *CVPR*. 2022.

[5] Minderer, M., et al. "Simple open-vocabulary object detection with vision transformers. *ECCV*. 2022.

**Questions:**

- In the abstract, the authors claim that "FSOD benchmarks are actually federated datasets", which is confusing. Can the authors explain it in detail?
- In Section 3.1, the author claimed "However, we argue that this setup is artificial, as it requires FSOD methods to detect car and person, among many other common categories given few-shot examples.", it would be best to explain it in detail in the paper.

---

### Official Review · Reviewer_PgvU · 2023-10-31

**Soundness:** 3 good
**Presentation:** 3 good
**Contribution:** 2 fair
**Rating:** 3
**Confidence:** 4

**Summary:**

This paper examines the use of foundational vision-language models for few-shot object detection tasks.  Using Detic as the pretrained base model, K-shot detection tasks simply fine-tune the model using the K box instances.  The use of pseudo-negatives and FedLoss variants help to better handle incomplete box annotations.  Ablations are also performed to look at the effect of freezing weights at various points (keeping the VLM-based classifier intact is most important).

**Strengths:**

Establishing a strong and relevant baseline for K-shot object detection is a good goal.  I fully agree with the argument that a realistic K-shot setting should make use of VLM foundational models.  This paper does just that, providing basic evaluations of Detic in few-shot transfer learning settings, providing a few nice improvements with pseudo-negatives, and useful measurements for layer freezing.

**Weaknesses:**

Overall, I find this work under-developed.  The idea of using Detic or other foundational model for few-shot object detection makes sense, but simply fine-tuning is only a small step away from the original Detic paper.  Indeed, the "zero-shot" setting is Detic as-is.  Fine-tuning beyond this, even with the additional negatives and frozen layers improvements, yields tangible gains, but not very sizable at about 1 AP.

While this is a good start, a more robust investigation into this idea could look into the effects of important factors like the size of the pretraining model, the size of its pretraining data, and the performance of few-shot detection on very different target dataset settings (e.g. medical scans, etc.).  This would provide a better picture of the method's behavior.

**Questions:**

Smaller questions:

* sec 4.4 "we run each K-shot experiment over 3 random data splits" --- What does this mean, exactly?  A straightforward reading is that for each class, 3 different random samples of images with K instances were chosen, so if there are 10 classes, 30 different few-shot training runs (each with K samples) are performed for the benchmark.  If this is what is intended, it seems like a small number; what is the variance (or max-min difference)?

* upper bound experiment:  I like the idea for the comparison, but I don't understand the description.  Do the exhaustive annotations exist in the nuImages dataset already?  If so, I'm not sure how images and instances were sampled for K-shot tasks: the original protocol described in sec. 3.2 says an image is chosen only if it has fewer than K instances so, so that the number of instances is exactly K.  It sounds like instead, a random set of (non-exhaustive) instances are chosen?

---

### Official Review · Reviewer_mJBQ · 2023-11-01

**Soundness:** 3 good
**Presentation:** 2 fair
**Contribution:** 2 fair
**Rating:** 5
**Confidence:** 5

**Summary:**

This work rethinks few-shot object detection with modern Vision-Language Models (VLM). Using stronger VLM foundation models, they achieves much stronger performance with few-shot fine-tuning than previous ImageNet pre-trained few-shot object detection models. During few-shot fine-tuning, in order to address the sparse annotation and smaller gradients for few-shot classes, they proposed the improved federated loss by generating pseudo-negatives. They evaluate their model on challenging LVIS and nuImages.

**Strengths:**

1. The idea of extending few-shot object detection (FSOD) with VLM foundation models is reasonable and interesting, which can potentially revolutionize the FSOD community.

2. The introduced federated loss is also sound, and a strong baseline for VLM foundation models based few-shot fine-tuning.

**Weaknesses:**

One major concern is the paper presentation regarding Section 3.3, which is the core technical contributions of this work. While in general, I can understand the challenges when doing few-shot fine-tuning with VLM models. I strongly suggest that the author rewrite this section to be more precise and formal, and introduce background knowledge about federated loss, and highlight their contributions and differences compared with the federated loss baseline.

Another concern is that the author uses vision-language pre-trained detection model (Detic) as the starting point which introduces much more bounding box annotations during pre-training compared with existing FSOD works, which may also include boxes of novel classes. Another way of introducing VLM foundation models like CLIP which does not include box bounding pre-training data seems to be more fair for evaluating few-shot object detection, like many of previous open-vocabulary object detection works [1]. Can the author provide more discussions about this?

[1] Gu, Xiuye, et al. "Open-vocabulary object detection via vision and language knowledge distillation." arXiv preprint arXiv:2104.13921 (2021).

**Questions:**

Please see the weaknesses above

---

### Official Review · Reviewer_C7qU · 2023-11-07

**Soundness:** 2 fair
**Presentation:** 3 good
**Contribution:** 2 fair
**Rating:** 3
**Confidence:** 3

**Summary:**

The authors mention that existing Few-Shot Objection Detection (FSOD) benchmarks are contrived and not representative of how FSOD is done in practice, i.e. it is claimed that large foundation models are downloaded and used either zero-shot or fine-tuned on the target dataset. This results in vocabulary overlap between train and test. The authors argue that benchmarks should be more closely aligned to this practical application and not artificially restrict concept leakage during pre-training.

Building on this premise, the authors observe that existing FSOD benchmark datasets closely resemble federated datasets (e.g. comprised of multiple subsets, where each subset guarantees exhaustive labelling). They present a simple strategy of altering the loss-function to account for not treating all non-detected instances as negatives due to non-exhaustive labelling for other categories.

Finally, the authors conduct experiments on LVIS and nuImages and show considerable improvement when fine-tuning a VLM with pseudo-negatives (instead of a regular loss or FedLoss)

**Strengths:**

The proposal to adapt various benchmarks when they no longer become useful or representative of the task being solved makes sense and should indeed be investigated

The authors make several observations that are useful for the larger community e.g. how best to handle FSOD datasets

**Weaknesses:**

The contribution seems pretty limited for a paper submission, especially since neither Detic nor FedLoss are a contribution of this paper and the InvFedLoss is a very small modification.

**Questions:**

Proposing that a VLM be used for FSOD may sometimes not be possible because of the large compute cost involved. This doesn't seem to be mentioned anywhere in the paper (both fine-tuning and inference) so it would be nice to get the authors' thoughts on a like-for-like comparison in terms of FLOPs.

Has there been any analysis on the text prompt for these tasks and how modifications of it affect the result.